# Prolyl Endopeptidase-Like Facilitates the α-Synuclein Aggregation Seeding, and This Effect Is Reverted by Serine Peptidase Inhibitor PMSF

**DOI:** 10.3390/biom10060962

**Published:** 2020-06-25

**Authors:** Gabriel S. Santos, William Y. Oyadomari, Elizangela A. Carvalho, Ricardo S. Torquato, Vitor Oliveira

**Affiliations:** 1Department of Biophisics, Escola Paulista de Medicina, Universidade Federal de São Paulo, São Paulo 04039-032, Brazil; santos.gabriel@unifesp.br (G.S.S.); williamoyadomari@yahoo.com.br (W.Y.O.); contatoelicarvalho@gmail.com (E.A.C.); 2Department of Biochemistry, Escola Paulista de Medicina, Universidade Federal de São Paulo, São Paulo 04039-032, Brazil; torquato@unifesp.br

**Keywords:** lewy body, amyloid fibrils, proteolysis

## Abstract

The aggregation of α-synuclein (α-Syn) is a characteristic of Parkinson’s disease (PD). α-Syn oligomerization/aggregation is accelerated by the serine peptidase, prolyl oligopeptidase (POP). Factors that affect POP conformation, including most of its inhibitors and an impairing mutation in its active site, influence the acceleration of α-Syn aggregation resulting from the interaction of these proteins. It is noteworthy, however, that α-Syn is not cleaved by POP. Prolyl endopeptidase-like (PREPL) protein is structurally related to the serine peptidases belonging to the POP family. Based on the α-Syn–POP studies and knowing that PREPL may contribute to the regulation of synaptic vesicle exocytosis, when this protein can encounter α-Syn, we investigated the α-Syn–PREPL interaction. The binding of these two human proteins was observed with an apparent affinity constant of about 5.7 μM and, as in the α-Syn assays with POP, the presence of PREPL accelerated the oligomerization/aggregation events, with no α-Syn cleavage. Furthermore, despite this lack of hydrolytic cleavage, the serine peptidase active site inhibitor phenylmethylsulfonyl fluoride (PMSF) abolished the enhancement of the α-Syn aggregation by PREPL. Therefore, given the attention to POP inhibitors as potential drugs to treat synucleinopathies, the present data point to PREPL as another potential target to be explored for this purpose.

## 1. Introduction

Synucleinopathy is a general term that refers to neurodegenerative diseases, including Parkinson’s disease (PD) [1,2], dementia with Lewy bodies (DLB) [1], diffuse Lewy body disease (DLBD) [3], and multiple system atrophy (MSA) [4,5]. The distinguishing characteristic of these neurodegenerative disorders is the presence of inclusions, in cells of the central nervous system, known as Lewy bodies (LBs), Lewy neurites (LNs), or glial cytoplasmic inclusions (GCIs) [6,7]. The major part of the protein fraction of these inclusions is formed by the accumulation of α-synuclein (α-Syn) aggregates [6,7]. When bound to membranes, α-Syn is structured; however, in solution, it is an intrinsically disordered protein that can suffer structural changes coupled with the formation of β-sheet rich α-Syn assemblies of various sizes [8,9,10,11], including the amyloid fibrils found in the LBs [12]. α-Syn is primarily an intracellular protein, and these β-sheet-rich α-Syn aggregates formed are thought to be toxic, leading to cell death [13,14]. The cell-to-cell spread of these β-sheet-rich α-Syn assemblies among neighbor cells is linked to the progression of these diseases, and the mechanisms of this event were the subject of various studies [15,16,17,18]. Despite the exhaustive investigation of α-Syn in these pathologies, the regular physiological functions of this protein are yet to be completely deciphered [7,19,20]. However, a general consensus on its cellular (neuronal) role can be cited; α-Syn is likely to have a regulatory function associated with the synapse (e.g., synaptic activity, synaptic plasticity, learning, neurotransmitter release, dopamine metabolism, synaptic vesicle pool maintenance, and/or vesicle trafficking) [9,19]. Furthermore, there is no connection between α-Syn aggregation and its regular function; on the contrary, this is only linked to pathologies. Therefore, one way to treat these synucleinopathies should be, exactly, to avoid the aggregation of this protein, which, as a matter of fact, is the main goal of several researchers [7], and it is also the main motivation of the present work. For that, we studied the interaction of α-Syn with prolyl endopeptidase-like (PREPL), an intracellular protein that also seems to play a role in pre-synaptic vesiculation and whose structure resembles prolyl oligopeptidase (POP) (instead of PREP (prolyl endopeptidase), we prefer to use POP (prolyl oligopeptidase) as a better distinction from the PREPL (prolyl endopeptidase-like) abbreviation in this text). Moreover, POP is known to interfere with the α-Syn aggregation [21].

Several studies showed that POP interacts with α-Syn and that this binding accelerates the characteristic α-Syn aggregation [22,23]. It was proposed that this POP–α-Syn complex serves as a seed for the aggregation process [23,24]. Interestingly, active site-directed POP inhibitors (e.g., KYP-2047 and Z-Pro-Prolinal) abolish the influence of this protein on α-Syn aggregation [22,23,25,26]. Moreover, the inactive POP Ser554Ala mutant also does not accelerate α-Syn aggregation (in vitro) [22], even though this mutant protein interacts with α-Syn with almost the same affinity as the POP wild type [23]. However, despite these observations, considering the main structural characteristics of POP, it is not possible to assume that α-Syn binds considerably (not in a substrate-like manner at least) at the POP active site. The POP active site is located between two domains, which excludes polypeptides larger than 30 residues as substrates [27]. In fact, α-Syn that has 140 residues is not cleaved by POP [22]. Additional observations clarified the link between the POP active site with its non-hydrolytic effect on the α-Syn. POP active site inhibitors, as well as this impairing Ser554Ala mutation, affect POP conformation [23,28,29,30], which, in turn, affect the resulting enhancement on the α-Syn aggregation presented by this peptidase. Other factors (protein–protein interactions or different environments) that can alter the POP conformational state or its compactness also influence its effect on the α-Syn aggregation event. For instance, this same inactive POP Ser554Ala mutant that does not accelerate the α-Syn aggregation (in vitro) can enhance the α-Syn dimerization in a cellular assay [23]. Taken together, the observations made so far show that different POP conformations can either induce or inhibit the aggregation of the αSyn [21]. This dual effect is the most attractive characteristic of the interaction of αSyn with POP. Additionally, POP inhibitors were shown to decrease α-Syn dimerization and increase autophagy in cells [26,31,32]. Autophagy is the most important cellular pathway to the degradation of aggregated proteins, and the mechanisms via which POP inhibitors induce this cellular response were studied [32,33,34]. Hence, considering all these in vitro and in vivo studies, POP inhibitors gained attention as potential drugs to treat synucleinopathies (e.g., Parkinson´s disease, PD) [26,31,35,36,37,38,39].

PREPL (prolyl endopeptidase-like) protein is structurally related to the proteins from clan SC of the serine peptidases, family S9, whose members are prolyl endopeptidase (PREP) or prolyl oligopeptidase (POP) (EC 3.4.21.26), oligopeptidase B (OPB) (EC 3.4.21.83), dipeptidylpeptidase IV (DPPIV) (EC 3.4.14.5), and acylaminoacyl-peptidase (APEH) (EC 3.4.19.1) [40]. However, despite the fact that the catalytic triad Ser–Asp–His directly involved in amide bond hydrolysis by these serine peptidases is also conserved in the PREPL structure (Ser559, Asp645, and His690) (Appendix A) [41], no measurable catalytic activity on peptide bonds could be detected in enzymatic PREPL assays so far [41,42,43,44]. On the other hand, the Ser residue is reactive toward PMSF (phenylmethylsulfonyl fluoride) [41] and DFP (di-isopropyl fluorophosphate) [41,42]; furthermore, PREPL can catalyze the hydrolysis of ester bonds [42]. Therefore, the enzymatic properties of PREPL remain to be better characterized.

Deletion of the *PREPL* (prolyl endopeptidase-like) gene was detected in patients with congenital myastemic syndrome 22 (CMS22) [45,46,47], and, together with the neighboring SLC3A1 gene, the *PREPL* gene is deleted in patients with hypotonia–cystinuria syndrome (HCS) [41,48,49]. At least seven different transcripts are potentially generated from two alternative transcription start sites present at the *PREPL* human gene structure [41]. Four of these transcripts codify to a shorter isoform of 638 amino-acid (aa) residues, and the other three transcripts can generate a protein with an 89-aa longer N-terminal, when translated (727 aa residues in total). PREPL is ubiquitously expressed; however, higher relative amounts of protein are detected in the brain, skeletal muscle, heart, and kidney [41,50]. More specifically, PREPL expression was already described in pyramidal neurons of the temporal cortex and neocortex [51], as almost all PREPL protein is probably contained in the cellular cytoplasm [41,51]. Studies of the patients with CMS22 suggest that PREPL may be important in the regulation of synaptic vesicle exocytosis, probably through association with other proteins also important for this cellular event [47].

Based on the α-Syn–POP studies and based on the above-mentioned PREPL properties, in the present work, we investigated if PREPL would also have a POP-like “non-hydrolytic” action on α-Syn aggregation. Thus, we examined (1) if PREPL also could interact with α-Syn, (2) if, as verified with POP, such an interaction would interfere with the α-Syn aggregation process, and (3) if an active site serine peptidase irreversible inhibitor (PMSF) could have any effect on the results.

Working with human PREPL (longer isoform of 727 aa residues) and α-Syn, we could detect the binding of these two proteins with an apparent affinity constant of about 5.7 μM and, as in the α-Syn aggregation kinetic assays with POP, the presence of PREPL accelerates the α-Syn oligomerization/aggregation process and, remarkably, the treatment with PMSF reduced the PREPL influence on the α-Syn oligomerization/aggregation.

## 2. Materials and Methods

### 2.1. Protein Expression and Purification

#### 2.1.1. α-Synuclein

A general cloning plasmid carrying the human wild-type α-Syn complementary DNA (cDNA) sequence was a kind gift from Dr. Jaap Broos (University of Groningen, Groningen, the Netherlands). The α-Syn coding region was excised from this plasmid and cloned into the pET26b vector with the *Nde*I and *Xho*I restriction enzymes (pET26b-α-Syn). After the verification of the correct in-frame cloning by nucleotide sequencing, *Escherichia coli (E.coli)* BL21(DE3) was subsequently transformed with this sequence verified pET26b-α-Syn. Protein expression procedure was initiated with a pre-culture, which was made from a single colony selected from an LB agar plate inoculated into 16 mL of LB medium and kept at 37 °C under 180 rpm shaking for overnight growing. The growth of the culture was continued by adding the pre-culture to one flask containing 800 mL of LB medium, which was kept at 37 °C under 220 rpm shaking until the absorbance measured at 600 nm reached 0.7. At this point, α-Syn expression was induced by the addition of 1 mM of isopropyl β-d-1-thiogalactopyranoside (IPTG) for 4 h. Colony selection, pre-culture, and culture were made with 50 μg/mL kanamycin in the medium. *E. coli* cells were then harvested by centrifugation at 8000× *g* for 30 min and subsequently resuspended in 20 mM Tris-HCl, pH 8.0 containing 1 mM PMSF and 5 mM EDTA. The lysis was performed as follows: the resuspended cells were kept in an ice bath and submitted to ultrasound pulses with 300 W for 30 min, but with alternating 15 s of pulses (15 s “on”) with intervals of 15 s without ultrasound pulses (15 s “off”). After cell lysis, the α-Syn purification was pursued by (1) acidification of the crude extract to precipitate contaminants, followed by centrifugation and neutralization of the supernatant, (2) ammonium sulfate precipitation of the supernatant, and (3) solubilization of the pellet from the ammonium sulfate precipitation step and dialysis against 50 mM Na_2_HPO_4_, pH 7.4 buffer containing 150 mM NaCl. The details of the α-Syn purification are described elsewhere [52]. Additionally, after dialysis, the purified α-Syn-containing sample was submitted to a polishing step, which consisted of a gel filtration chromatography in a Superdex 200 10/300 GL column (GE Healthcare, Marlborough, MA, USA) at a flow rate of 0.5 mL/min in the same dialysis buffer (50 mM Na_2_HPO_4_, pH 7.4 buffer containing 150 mM NaCl), when only the fractions corresponding to the monomeric α-Syn retention times were collected. The protein concentrations were determined by absorbance measurements at 280 nm, using the molar extinction coefficient ε_α-Syn_ = 5600 M^−1^·cm^−1^. The α-Syn monomeric fractions were characterized by SDS/PAGE and circular dichroism (CD).

#### 2.1.2. PREPL and POP

The codon-optimized (for *E. coli*) cDNA coding sequences for human PREPL and POP were synthetized (Genscript, Piscataway, NJ, EUA) and cloned into pET26b vector between *Nde*I and *Xho*I restriction sites, resulting the two constructs: pET26b-PREPL and pET26b-POP vectors. After the verification of the correct in-frame cloning by nucleotide sequencing, *E. coli* BL21(DE3) samples were subsequently transformed with the sequence verified pET26b-PREPL or the pET26b-POP vector. Protein expression, including cell disruption by sonication, followed the same procedures as described above for the α-Syn expression, but with a few differences; the IPTG induction step was longer (16 h) and at a lower temperature of 20 °C, while the cells were resuspended for cell disruption in a different buffer, i.e., 20 mM Tris-HCl, pH 7.4 containing 500 mM NaCl, 4 mM imidazole and 0.2 mg/mL lysozyme. After cell lysis, the crude cell extract was centrifuged at 8000× *g* for 30 min and the supernatant was collected. The cleared lysates were then purified on an AKTA purifier system (GE Healthcare) coupled with a Histrap FF column. The column-bound proteins were eluted with increasing imidazole concentrations. PREPL (or POP) fractions eluted with approximately 100 mM of imidazole. The eluted samples were pooled and concentrated in an Amicon^®^ ultra 10 kDa cutoff (Merck, Darmstadt, Germany) and then applied to a Superdex 200 10/300 GL column (GE Healthcare) at a flow rate of 0.5 mL/min in 50 mM Na_2_HPO_4_, pH 7.4 buffer containing 150 mM NaCl. The PREPL (or POP) fractions were characterized by enzymatic activity (MUGB substrate for PREPL and Z-Gly-Pro-MCA substrate for POP), SDS/PAGE, and CD.

The protein concentrations were determined by absorbance measurements at 280 nm, using the following molar extinction coefficients: ε_PREPL_ = 93,170 M^−1^·cm^−1^ and ε_POP_ = 128,090 M^−1^·cm^−1^. Aliquots were frozen stored at −80 °C with 10% (*v*/*v*) glycerol.

### 2.2. Activity Assays

#### 2.2.1. PREPL Activity Assay with MUGB

4-Guanidinobenzoic acid 4-methylumbelliferyl ester hydrochloride (MUGB) was used as substrate for PREPL. For the activity measurement, 0.25 mM MUGB was incubated with 1.5 µM PREPL in 50 mM Na_2_HPO_4_, pH 7.4 buffer containing 150 mM NaCl. Inhibition tests were made by adding 1 mM PMSF, 0.145 mM KY-2047, or the respective vehicles to the reactions, 2 min prior to the substrate addition. Reactions were prepared in clear-bottom 96-well plates with a final reaction volume of 200 μL in each well. The samples were analyzed in a multiplate reader (Synergy H1, BioTek, Winooski, VT, USA) set with 320 nm and 460 nm excitation and emission wavelengths, respectively. Fluorescence data were collected every 55 s for 20 min under continuous orbital shaking at 37 °C.

#### 2.2.2. POP Activity Assay with Z-GP-MCA

Z-GP-MCA was used as substrate for POP. For the activity measurement, 10 μM of this compound was incubated with POP samples (10 nM) in 20 mM Tris-HCl, pH 8.0 containing 150 mM NaCl. Reactions were prepared in 2-mL quartz cuvettes and continuously stirred at 37 °C. The product formation was followed by the continuous fluorescence intensity measurement for 1 h in a spectrofluorometer HITACHI F-2500 (HITACHI, Tokyo, Japan) (set with λ_EX_ = 380 nm and λ_EM_ = 460 nm).

#### 2.2.3. Tests to Detect PREPL Hydrolytic Activity on Amide Bonds

Z-FR-MCA, Z-GP-MCA, and Abz-GFSPFRQ-EDDnp were tested as PREPL substrates. Each compound (10 μM) was incubated separately with increasing PREPL concentrations (until 0.1 μM) in 20 mM Tris-HCl, pH 8.0 containing 150 mM NaCl at 37 °C. The reactions were followed by continuous fluorescence intensity measurement for 1 h in a spectrofluorometer HITACHI F-2500 (λ_EX_ = 380 nm, λ_EX_ = 460 nm for the MCA substrates and λ_EX_ = 320 nm, λ_EM_ = 420 nm for the Abz substrate). As any activity was detected in the continuous activity assays, prolonged incubations of 24 h were tried. Aliquots were taken at 1 h, 2 h, 8 h, and 24 h and analyzed by HPLC. α-Syn (1 mM) and bradykinin (10 μM) were also incubated for 24 h with PREPL with aliquots taken and analyzed by HPLC as well.

Analytical HPLC was done using a binary pump system from Shimadzu (Shimadzu, Kyoto, Japan) fitted with an ultraviolet–visible light (UV–Vis) detector. The system was coupled to a C4 column (5 µm, 4.6 mm × 150 mm) that was eluted with solvents: A (water/trifluoracetic acid at a proportion of 99:1) and B (water/acetonitrile/trifluoracetic at a proportion of 89:10:1). The elution performed at a flow rate of 1 mL/min and a 10–80% gradient of B over 20 min was monitored continuously by the absorbance at both 220 nm and 280 nm.

### 2.3. Circular Dichroism (CD)

CD spectra were carried out on an Applied Photophysics Chirascan Plus spectropolarimeter (Applied Photophysics, Surrey, UK). The parameters were as follows: sensitivity 100 mdeg; resolution of 0.5 nm; response time 4 s; scan rate 20 nm/min. Each recorded spectrum is an average of four sequential scans at 25 °C. The control baseline was obtained with all buffer components (Tris-HCl 20 mM, pH 7.4 with NaCl 150 mM) prepared without the protein. The spectra of monomeric α-Syn (2 µM, 4 µM, or 8 µM) alone or mixed with PREPL (2 μM) were collected in the far-UV range (190–250 nm) using a 0.2-mm-path-length quartz cell. The α-Syn spectra were analyzed by using the online BeStSel web server tool [53], whereas the PREPL spectra were compared with the spectra also obtained for human POP and porcine POP (Appendix A).

### 2.4. Surface Plasmon Resonance (SPR)

Binding studies were performed using a Biacore T200 system (GE Healthcare) equipped with a clean sensor chip CM5 (carboxymethylated dextran matrix, GE Healthcare). For immobilization of the proteins in the sensor chip surface, PREPL (20 mg/mL) or POP (40 mg/mL) samples were injected in 5 mM acetate pH 4.0 or pH 4.5, respectively, over the chip surface pre-activated with a mixture of l-ethyl-3-(3-dimethylaminopropyl) carbodiimide (EDC) (20 mg/mL)/*N*-hydroxysuccinimide (NHS) (5 mg/mL) solution. The surface was then washed with the HBS + EP buffer (0.01 M HEPES, pH 7.4, 0.15 M NaCl, 3 mM EDTA, 0.005% *v*/*v* Surfactant P20) and subsequently deactivated with 1 M ethanolamine, pH 8.5 solution. The immobilization resulted with 1134.6 response units (RU) for PREPL and 1514.1 RU for POP. The specificity of both PREPL and POP was tested with injections of bovine serum albumin (BSA) as analyte. No unspecific binding was detected. To analyze the interaction between α-Syn and the immobilized proteins (PREPL or POP), buffer HBS + EP was used. The analyte α-Syn was diluted to various concentrations (from 0.1 to 250 µM) in the same buffer and injected for 350 s during the association phase at a constant flow rate of 15 µL/min at 25 °C. The subsequent dissociation phase was carried out for 1000 s at the same flow rate. The surface of the sensor chip was regenerated using 2 M NaCl and 2 M glycine, pH 2.0 for 30 s. Binding affinity was determined by plotting α-Syn concentration vs. resonance units and calculated using steady-state methods. Binding affinity was determined by plotting α-Syn concentration vs. response (resonance units) and calculated using the one-site ligand binding equation,
*Response* = (*Response_max_.*[*α* − Syn])/(*K_d_* + [*α* − *Syn*]),(1)
and with the initial estimates provided by use of linear fitting utilizing the Scatchard rearrangement, using Grafit 5.0 (Eritacus software, West Sussex, UK). *K_d_* (binding affinity constant) was also calculated using the steady-state parameters *ka* (association constant) and *kd* (dissociation constant) provided by the Biacore T200 Evaluation Software version 1.0 (GE Healthcare).

### 2.5. α-Syn Aggregation Assays

#### Thioflavin T (ThT) Fluorescence

The time course of human recombinant α-Syn aggregation was followed by measuring of the thioflavin (ThT) fluorescence intensity change in a multiplate reader (Synergy H1, BioTek, USA). α-Syn (150 μM) alone or mixed with PREPL (60 nM, 120 nM and 240 nM) or POP (120 nM) was kept in 50 mM NaH_2_PO_4_ buffer, pH 7.4 containing 150 mM NaCl and 8 μM ThT at 37 °C for 24 h under double-orbital shaking. When appropriate, the aliquots of PREPL or POP were pre-incubated with an excess of PMSF (>1 mM) or KYP-2047 (1 μM) prior to the addition to the final mixture. The ThT fluorescent assays were done in clear-bottom 96-well plates (Thermo Scientific, Waltham, MA, USA,) with a final volume of 100 μL per well. The plates were sealed with clear polyolefin sealing tape. To each well, an 1/8” diameter glass ball was added. Bottom fluorescence intensity (λ_EX_ = 450 nm, λ_EM_ = 477 nm) was recorded at 30-min intervals. Experiments were conducted in triplicate in three independent experiments. Fluorescence over time data were fitted to a sigmoidal equation,
*y* = *y*_0_ + (*y_max_* − *y*_0_)/(1 + *exp*(−*k_app_*(*t* − *t*_1/2_)),(2)
using Grafit 5.0 (Eritacus software), where *y* is the ThT fluorescence at a particular time point, while *y_max_* and *y_0_* are the initial and maximum ThT fluorescence, respectively. The lag time values were then calculated from the *t*_1*/*2_ and *k_app_* obtained parameters as follows [54]:(*lag time* = *t*_1/2_ − *2/k_app_*).(3)

## 3. Results and Discussion

### 3.1. PREPL Is Active toward an Ester Substrate (MUGB), and It Is Inhibited by PMSF but Not by KYP-2047

Because of its homology with the peptidases from the S9 family, PREPL was expected to hydrolyze peptide bonds following arginine residues (Appendix A) [41]. However, the purified recombinant human PREPL we obtained was only able to hydrolyze the ester bond of the MUGB substrate (Figure 1). Even with prolonged incubations (24 h), and with relatively large PREPL concentrations, the fluorogenic substrates Z-FR-MCA and Z-RR-MCA (substrates with the Arg residue at their P_1_ position and known to be hydrolyzed by OPB) were not cleaved. On the other hand, despite this expected PREPL hydrolytic activity on peptide (amide) bonds, the PREPL enzymatic activity pattern verified actually supports previous works [42]. Further supporting previously published data, we observed that PREPL was irreversibly inhibited by PMSF (Figure 1), which indicates that a very reactive serine residue is present in the PREPL structure [44], which is probably the Ser559 residue of the catalytic triad conserved in family S9 (Appendix A). Nonetheless, the identification of the enzymatic activity of PREPL as a peptidase remains to be elucidated. Nevertheless, as in the present work it was investigated if PREPL would also have a POP-like “non-hydrolytic” action on α-Syn aggregation, the following further observations are noteworthy: even with prolonged incubations (24 h), and with relatively large PREPL concentrations, neither Z-GP-MCA and bradykinin (POP susceptible substrates) nor α-Syn were cleaved by PREPL. Furthermore, the active site POP inhibitor KYP-2047 did not inhibit the PREPL activity on MUGB (Figure 1). This last observation is noteworthy because, beyond blocking the POP enzymatic activity (in vitro and in vivo) [55], KYP-2047 also prevents the observed POP enhancement of the α-Syn oligomerization/aggregation processes, showing many beneficial effects, in studies with animals, preventing α-Syn aggregation and even inducing the α-Syn aggregates clearance [26,31]. Therefore, these results confirm that PREPL is not a KYP-2047 target.

### 3.2. PREPL Does Not Cleave but Interacts with α-Syn

Far-UV CD assays helped in the characterization of the purified proteins; however, most importantly, using this analysis, the direct protein–protein interaction between PREPL and α-Syn could be initially detected.

α-Syn spectra obtained are typical spectra of structurally disordered proteins (Figure 2). The spectra deconvolutions indicated at least 60% random structures. The CD analyses of samples with increasing α-Syn concentrations resulted in identical spectra after normalization by the protein concentration, indicating that the prepared samples did not contain considerable amounts of aggregated α-Syn, being mostly formed of α-Syn in its soluble and monomeric form (data not shown). Far-UV CD spectra of the PREPL were compared with the spectra obtained for POP, for which a crystallographic structure is already known. This comparison of the CD spectra obtained for PREPL and POP (human and porcine) is shown in Appendix A, and, considering the experimental errors, the CD spectra of these two proteins are quite similar (for more details about this comparative structural analysis, please refer to comments on Appendix A in the Appendix A).

Changes in the UV CD spectra obtained before and after addition of α-Syn to PREPL clearly show the binding between these two proteins (Figure 2). If no binding occurred, both spectra (α-Syn alone or mixed with PREPL) represented by the red and blue lines would be identical. A possible interpretation for the changes visualized in this CD analysis can be made assuming that, as the α-Syn is an intrinsically disordered protein, while PREPL is a well-folded large protein, the variations of the UV CD spectra may mainly represent changes of the α-Syn structure. However, it cannot be discarded that PREPL may have also structural modifications upon α-Syn binding, which also account for the spectra differences shown in Figure 2. Nevertheless, although no quantitative data can be taken from these spectra obtained from PREPL+ α-Syn mixtures, it is quite clear that a conformational change (including secondary structural variations) is coupled with the binding of these two proteins.

Surface plasmon resonance (SPR) assays complemented the α-Syn–PREPL binding study (Figure 3). Titrations of immobilized PREPL with increasing α-Syn concentrations resulted in an apparent binding constant (*K_d_*) of 5.7 ± 0.5 μM) when the dose–response data were fitted to Equation (1). By using the steady-state parameters (*k_a_* = 719 ± 2 M^−1^·s^−1^ and *k_d_* = 0.001025 ± 0.000003 s^−1^), the binding constant was determined to be about 1.4 × 10^−6^ (~1.4 μM). This affinity is very similar to the binding constant observed for α-Syn binding to immobilized POP by Savolainen et al. in SPR assays of about 3.61 μM [23].

It is noteworthy, however, that the injections with higher concentrations (>100 μM) of α-Syn provoked a further strong response increase. We interpreted this occurrence as the oligomerization of the α-Syn on the sensorchip surface and, therefore, we worked only with α-Syn concentrations lower than 50 μM, when the response was predominantly given by 1:1 binding, which was verified by checking the linearity of the corresponding Scatchard plots at the different concentration ranges. This phenomenon was even more accentuated in the titrations of POP immobilized at the sensorchip with increasing α-Syn concentrations. As we could not observe a dose–response curve for the titrations with POP immobilized, probably because the affinity of the α-Syn for POP is higher (Appendix A), this may have caused the α-Syn oligomerization on the sensor chip surface with POP immobilized at lower concentrations than the observed in the SPR assays with PREPL immobilized. The adjusted sensorgrams obtained in the titrations of POP immobilized at the sensorchip with increasing α-Syn concentrations are presented in the Appendix A.

### 3.3. PREPL Accelerates the α-Syn Oligomerization/Aggregation, and This Effect Is Inhibited by PMSF

The results presented in Figure 4 show that α-Syn aggregates faster in the presence of PREPL. Figure 4A shows the time course of the α-Syn aggregation, followed by the increasing of amyloid-bound ThT fluorescence, in the absence and in the presence of increasing concentrations of PREPL. Each obtained dataset was fitted to Equation (2) and the parameters (*k_app_* and lag time) obtained were plotted against the corresponding concentration of PREPL present in the aggregation assay, as shown in Figure 4B,C. Increasing PREPL concentrations (60 nM, 120 nM, and 240 nM) caused a proportional decrease in the lag times (Figure 4B) and a concomitant proportional increase in the growth rate constants (Figure 4C) of the time course of the α-Syn aggregation. 

To test if, as observed with POP, the PREPL inhibition would also affect its influence on the α-Syn aggregation, ThT assays were carried out with PREPL treated with the irreversible active site serine protease inhibitor PMSF. For these assays, a fixed PREPL concentration of 120 nM was used. Comparative assays with POP untreated or treated with PMSF or KYP-2047 were also carried out. The results of these tests are summarized in Table 1.

The obtained data show that the treatment of the proteins with PMSF interfered with the enhancement on α-Syn aggregation observed in the presence of PREPL and POP. It is noteworthy that the PMSF not only blocked the enhancement effect of both PREPL and POP on the α-Syn aggregation, but it can also be observed that, in the PREPL/PMSF-treated or POP/PMSF-treated assays, the aggregation of α-Syn was even slower than in the respective control assay (α-Syn + PMSF) (Table 1), indicating that both POP/PMSF- and PREPL/PMSF-inactivated enzymes bind to α-Syn, as well as the untreated active proteins. By activity measurements (Z-FR-MCA substrate for POP and MUGB substrate for PREPL), we verified that the amount of PMSF added to the assays was enough to completely inhibit the activity of these enzymes. It is important to note that the PMSF does not react non-specifically with serine residues, but it only reacts with the catalytic serine residues of the serine peptidase active sites due to the exacerbated reactivity of such serine residues. These results with the PMSF-treated proteins are in accordance with previously published works showing that POP inhibitors block the effect of this protein on the α-Syn aggregation [23,26,31,35], observations that we confirmed, in our assay conditions, by performing an additional control experiment with POP treated with one of these specific inhibitors, KYP-2047 (Table 1). These results also indicate that, as also observed in previous works with POP [22,23,24], the α-Syn binding to an enzymatically active PREPL is crucial for the α-Syn oligomerization/aggregation enhancement, despite the fact that α-Syn is not a substrate for PREPL or for POP [22]. As mentioned in the introductory section, the observations made so far show that different POP conformations can either induce or decrease the aggregation of the α-Syn [21,23]. POP active site inhibitors affect the POP conformational state or its compactness [23,28,29,30], which, in turn, modulate the resulting enhancement on the α-Syn aggregation presented by this peptidase. Similarly, a PREPL conformational change, caused by the PMSF treatment, is the most plausible explanation for the observed opposite effect on the aggregation of the α-Syn, where the results obtained with the untreated PREPL are compared with those obtained with the PMSF-treated PREPL (Table 1).

## 4. Conclusions

In vivo assays aiming to further support the PREPL role on the α-Syn aggregation are in progress. Even so, taking the PREPL expression, its cellular localization pattern, and the strong parallel with POP, concerning the interaction with α-Syn, the results described in the present work put PREPL as another target for drugs aiming to treat synucleinopathies (e.g., Parkinson’s disease). Efforts to develop PREPL inhibitors were already made by other researchers [44]; consequently, the observations published here can motivate the usage of these obtained compounds to this end, including the possibility that the association of POP and PREPL inhibitors may improve the beneficial effects on PD treatment tests observed so far with POP inhibitors alone [26,31,36,38].

## Figures and Tables

**Figure 1 biomolecules-10-00962-f001:**
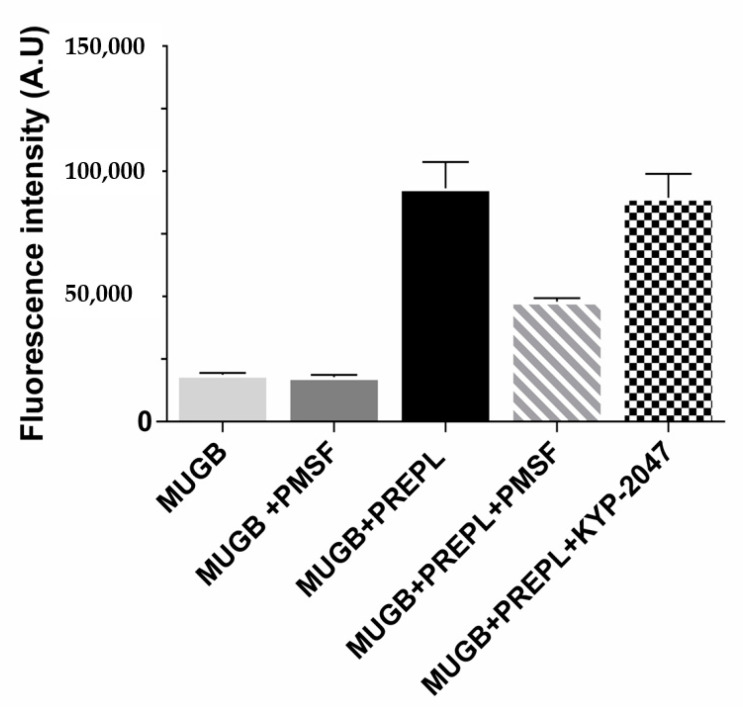
Prolyl endopeptidase-like (PREPL) enzymatic activity assays using 4-guanidinobenzoic acid 4-methylumbelliferyl ester hydrochloride (MUGB) as substrate. Fluorescence intensity (λ_EX_ = 320 nm, λ_EM_ = 460 nm) increased with MUGB ester bond hydrolysis. MUGB hydrolysis by PREPL was inhibited by phenylmethylsulfonyl fluoride (PMSF) (50%) but not by KYP-2047.

**Figure 2 biomolecules-10-00962-f002:**
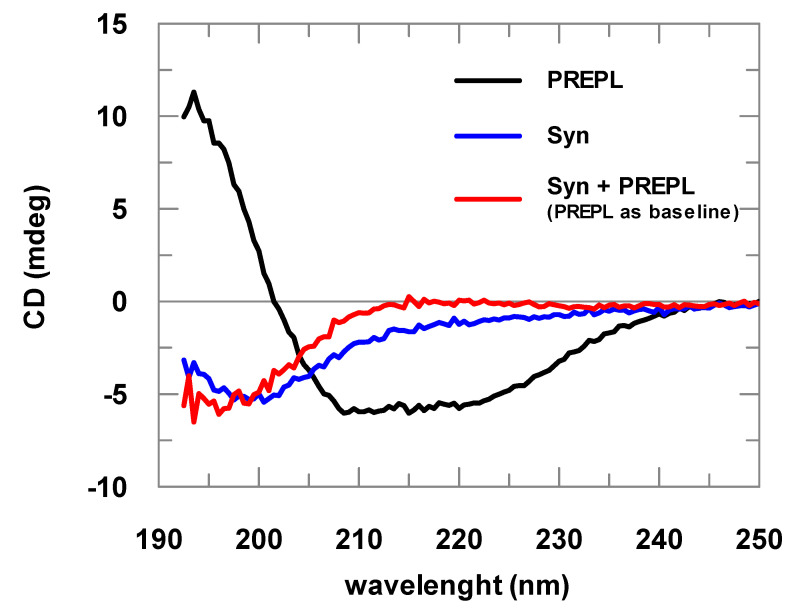
Far-ultraviolet (UV) circular dichroism (CD) spectra of PREPL (2 μM), α-synuclein (α-Syn; 8 μM), and PREPL (2 μM) mixed with α-Syn (8 μM). The different α-Syn spectra obtained in the absence of PREPL (blue line) and in the presence of PREPL (red line) show the interaction between these proteins, where a conformational change is coupled with α-Syn binding to PREPL.

**Figure 3 biomolecules-10-00962-f003:**
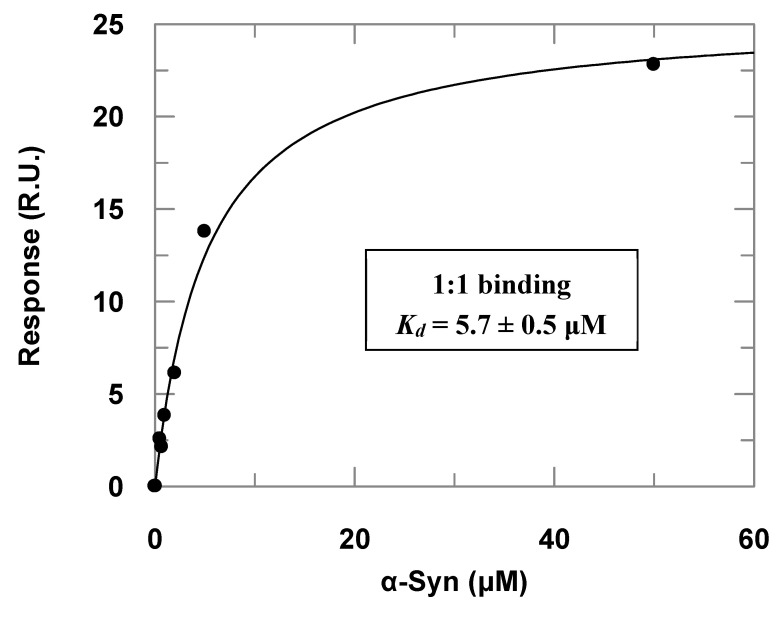
Dose–response data obtained in the surface plasmon resonance (SPR) assays with immobilized PREPL and α-Syn as the analyte injected at increasing concentrations (the correspondent adjusted sensorgrams are presented in the Appendix A). The symbols represent the response in resonance units (RU) determined experimentally at each concentration of injected α-Syn, and the line is a fitted curve according to Equation (1).

**Figure 4 biomolecules-10-00962-f004:**
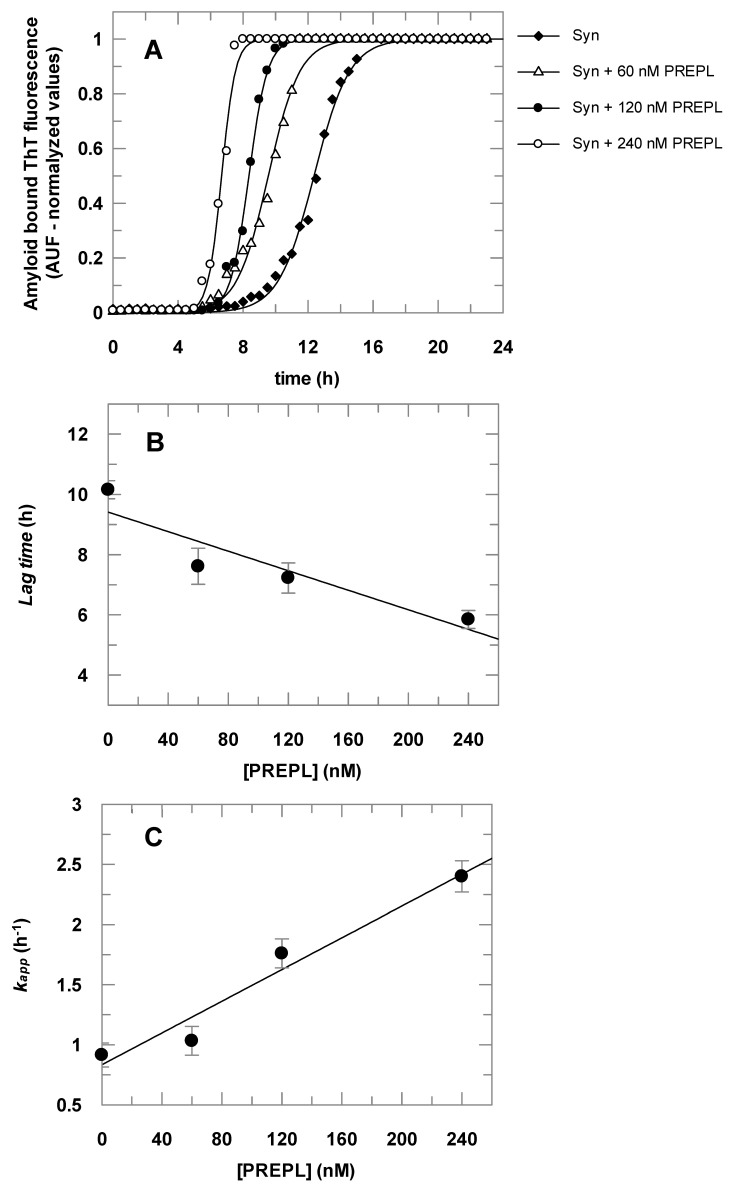
Effect of PREPL on the α-Syn aggregation. **Panel A**: The α-Syn amyloid fibril formation was monitored by thioflavin T (ThT) fluorescence for 24 h. The assays were carried out in the absence and in the presence of increasing PREPL concentrations (60 nM, 120 nM, and 240 nM). Data symbols are shown in the right upper side legend. The symbols represent ThT fluorescence intensities determined experimentally, and the lines are fitted sigmoidal curves according to Equation (2). Panels B and C: Lag times and growth rate constants (*k_app_*) obtained from the data fitting (results from the assay represented in Panel A) were plotted according to the concentration of PREPL present at the corresponding α-Syn aggregation assay. **Panel B**: Lag time versus PREPL concentration; **Panel C**: Growth rate constant versus PREPL concentration. Panel A shows representative data obtained in one assay, whereas data in Panels B and C represent the average values with the respective standard deviations obtained in three independent assays.

**Table 1 biomolecules-10-00962-t001:** Growth rate constant and lag time parameters for α-Syn fibril formation, monitored by ThT fluorescence over time, in the absence or in the presence of PREPL and POP (− or + PMSF and − or KYP-2047).

Sample and Condition	*k_app_* (h^−1^)	Lag Time (h)
α-Syn	1.18 ± 0.06	10.3 ± 0.3
α-Syn + PMSF	1.11 ± 0.09	10.1 ± 0.5
α-Syn + KYP-2047	1.2 ± 0.1	9.9 ± 0.5
α-Syn + PREPL^a^	1.76 ± 0.06	7.4 ± 0.5
α-Syn + PREPL^a^ + PMSF	1.2 ± 0.1	15.7 ± 0.5
α-Syn + POP^a^	1.7 ± 0.1	7.7 ± 0.7
α-Syn + POP^a^ + PMSF	1.0 ± 0.1	13.2 ± 0.4
α-Syn + POP^a^ + KYP-2047	1.2 ± 0.1	12.2 ± 0.5

^a^[PREPL] = [POP] = 120 nM.

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
