# Peer review of "Prolyl Endopeptidase-Like Facilitates the α-Synuclein Aggregation Seeding, and This Effect Is Reverted by Serine Peptidase Inhibitor PMSF"

_biomolecules, 2020, doi:10.3390/biom10060962_

Round 1
Reviewer 1 Report
In this study, authors investigated Prolyl endopeptidase-like (PREPL) protein interaction with with α-Synuclein. Using in vitro approaches authors also reported PREPL accelerates the α-Syn oligomerization-aggregation, However, neither literature or previously published data, nor those provide by the authors, justufy the objective of studing the interaction of PREPL and a-synuclein. There is nothing to justify its potential role in Parkinson´s disease pathology or in in vivo a-synluclein aggregation.
Reviewer 2 Report
Attached.

Reviewer 3 Report
Review of the manuscript “Prolyl endopeptidase-like facilitates the α-synuclein aggregation seeding, and this effect is reverted the by serine peptidase inhibitor PMSF” by Gabriel S. Santos and coauthors submitted to “Biomolecules”.
Neurodegenerative disease are devastating disorders an efficient treatment for which is not yet found. Synucleinopathies are one of the important group of the neurodegenerative disorders, so investigation of synuclein as a key component in the pathology is in the focus of many investigators. The study of prolyl oligopeptidase and alpha-synuclein are important, since they may lead to the development of treatment of these diseases. The manuscript contains new important data which will be interesting for the readership of “Biomolecules”
The following corrections should be done.
Line 18: “Being that one of the most intriguing questions about PREPL is exactly, its apparent lack of peptidasic activity”. This is a clumsy sentence. May be the authors want to say: ”One of the most intriguing questions about PREPL is its apparent lack of peptidasic activity”?
Line 21:”… when this protein can encounter α-Syn, in this work we investigated the α-Syn-PREPL interaction. The binding of these” The authors should delete “in this work”
Line 26: “Therefore, given the attention gathered by POP inhibitors as potential drugs to treat synucleinopathies, the present data put PREPL as another target to be explored for this purpose”. This is an awkward sentence which should be rewritten as follows: ”Therefore, given the attention to POP inhibitors as potential drugs to treat synucleinopathies, the present data point to PREPL as another potential target to be explored for this purpose”.
Lines 31-35 “The accumulation of protein aggregates mostly formed by α-synuclein (α-Syn) [1], known as Lewy bodies (LBs), in the central nervous system, is a distinguish characteristic of neurodegenerative diseases termed synucleinopathies [1,2], which include: Parkinson´s disease (PD) [3]; dementia with Lewy bodies (DLB) [3]; diffuse Lewy body disease (DLBD) [4] and multiple system atrophy (MSA) [5,6]”
This long sentence should be split into two to become more reader friendly. After reference [3] the authors should add a more recent review on PD: Emamzadeh et al., “Parkinson’s disease: Biomarkers, Treatment, and Risk Factors. Front. Neurosci., 30 August 2018, 12:612 | https://doi.org/10.3389/fnins.2018.00612.
Lines 36-37 “…the formation of β-sheet rich α-Syn assemblies of various sizes [7–9], including the amyloid fibrils found in the LBs” The authors should add a citation of the following “New alpha- and gamma-Synuclein Immunopathological Lesions in Human Brain. Acta Neuropathologica Com, 2014, 2: 132”.
Line 41 “Despite the exhaustive investigation of α-Syn in these pathologies, the regular physiological function(s) of this protein has(have) not been completely deciphered yet [2,17].”
The authors should add a citation of a recent review on α-Syn “Intracellular dynamics of synucleins: Here, there and everywhere. International Review of Cell Molecular Biology, 2015; 320:103-169.
Materials and Methods
Lines 101-102 “A general cloning plasmid carrying the human wild-type α-Syn gene was a kind gift from
Jaap Broos (University of Groningen, Groningen, The Netherland). The human wild-type α-Syn gene
wild-type α-Syn…”
This statements sound confusing. α-Syn gene contains introns and non-coding sequences. If the authors used only coding parts of the gene, they should say α-Syn cDNA.
Line 109 “The culture was continued by…” should be written as “The growth of the culture was continued…”
Lines 107-110 there is no need to repeat three times “50 μg/ml kanamycin”, the explanation can be given once.
Line 132 “Synthetic and codon optimized (for E. coli) human PREPL and POP genes (Genscript, 132) see the comments to α-Syn gene above (lines 101-102).
Chapters 2.1.1 and 2.1.2 are overloaded with details which describe routine, standard methods. May be truncated and replaced by references to published articles.
Citation. The authors should quote an important recent relevant review by Svarcbahs and coauthors: "New tricks of prolyl oligopeptidase inhibitors - A common drug therapy for several neurodegenerative diseases". Biochem Pharmacol. 2019 Mar;161:113-120.
Reviewer 4 Report
In 2008, Brandt et al. showed in vitro that a serine protease, prolyl oligopeptidase (POP, PREP) can enhance the aggregation of alpha-synuclein and this can be blocked by a specific POP inhibitor. Later, several papers have shown that POP inhibitors can reduce alpha-syn aggregation also in cellular and animal models. In this paper, Santos et al. present an interesting finding showing that PREPL enzyme facilitates alpha-syn aggregation in vitro in similar interaction-based manner as POP. This could be an interesting starting point for PREPL-alpha-syn studies, similar to Brandt et al. was for POP-alpha-syn studies. However, I have some comments and suggestions.
In addition to interaction studies, Thioflavin T assay showing that PREPL accelerates alpha-syn aggregation is the most important assay and result in this article. Although MST assay is done well, I have some concerns regarding the ThT assay:
- Why time was only 24h? Generally alpha-syn aggregation has a lag phase of 12-36 h in aggregation assays. Did the authors sonicate alpha-syn before initiating ThT aggregation assay? Monomeric alpha-syn aggregates easily even during the storage, and if e.g. oligomers were present in the alpha-syn solution, this can explain much faster aggregation in ThT assay than usually. This should be clarified.
- Authors should have used variable concentrations of PREPL in the ThT assay, and also variable concentrations of PMSF to show dose-response.
- In POP-alpha-syn ThT assay, a specific POP inhibitor, such as KYP-2047, in comparison with PMSF would be informative.
- Figure 4B should be also shown with line graph as in Figure 4A. I also don’t understand the difference between the information presented in Figure 4B and Table 1, please clarify.
Other comments:
In abstract and introduction, authors discuss about the nature of interaction with POP and alpha-syn (abstract “Interestingly, for this accelerated aggregation, α-Syn must bind to POP active-site…” and introduction “Moreover, the inactive POP Ser554Ala mutant also does not accelerate the α-Syn aggregation (in vitro) [19], even though this mutant protein interacts with α-Syn with almost the same affinity than the POP wild type [20]”). It is far from clear that alpha-syn would bind to active site of POP to achieve this, alpha-syn should be able to find a way inside POP to active site. It is more likely an interaction between alpha-syn and certain POP conformations than POP active site (as discussed in Savolainen et al. 2015). Authors also omit the fact that Ser554 mutated catalytically inactive POP induces alpha-syn aggregation in a cellular model (not in vitro though) that further underlines the active site is not needed for alpha-syn aggregation. There is also a very recent paper by Kilpeläinen et al. (Biomed Pharmacother. 2020) showing that strong enzymatic inhibition of POP by a small-molecule inhibitor is not required for alpha-syn effects. Authors should correct these claims in the manuscript.
Introduction:
Line 39: “Even though, before or during cell death these β-sheet rich α-Syn assemblies may be segregated, probably by an autophagy linked process, and propagated among neighbor cells [13–16].” I think this is oversimplification about alpha-syn propagation process as it is not clear how alpha-syn fibrils exit cell. Additionally, the effect of autophagy inducers on alpha-syn propagation can be totally different than what the authors claim in the manuscript (e.g. Kim et al. Autophagy 2016, 12(10):1849-1863.
doi: 10.1080/15548627.2016.1207014).
Round 2
Reviewer 1 Report
Authors provided a satisfactory justification of the objective of studing the interaction of PREPL and a-synuclein in Parlinson's disease. They support this in vitro proof-of-principle study as a new potential molecular target for Parkinson’s Disease, and it fits to what is expected from a “Communication” in the special issue within Biomolecules of the Biochemistry section: Potential Molecular Targets for Disease—Modifying Therapeutic Strategies in Parkinson’s Disease.
Reviewer 4 Report
My comments have been taken care of and the manuscript can be accepted.